# Hitchhiking Mapping of Candidate Regions Associated with Fat Deposition in Iranian Thin and Fat Tail Sheep Breeds Suggests New Insights into Molecular Aspects of Fat Tail Selection

**DOI:** 10.3390/ani12111423

**Published:** 2022-05-31

**Authors:** Mohammad Hossein Moradi, Ardeshir Nejati-Javaremi, Mohammad Moradi-Shahrbabak, Ken G. Dodds, Rudiger Brauning, John C. McEwan

**Affiliations:** 1Department of Animal Science, Faculty of Agriculture, Arak University, Arak 38156-8-8349, Iran; 2Department of Animal Science, University of Tehran, Karaj 31587-1-1167, Iran; ardeshir.nejati@gmail.com (A.N.-J.); moradim@ut.ac.ir (M.M.-S.); 3Centre for Reproduction and Genomics, AgResearch, Invermay, Mosgiel 9053, New Zealand; ken.dodds@agresearch.co.nz (K.G.D.); rudiger.brauning@agresearch.co.nz (R.B.); john.mcewan@agresearch.co.nz (J.C.M.)

**Keywords:** genomic scan, selection signature, lipid metabolisms, candidate genes, fat tail sheep

## Abstract

**Simple Summary:**

Fatness-related traits are economically very important in sheep production and are associated with serious diseases in humans. Using a denser set of SNP markers and a variety of statistical approaches, our results were able to refine the regions associated with fat deposition and to suggest new insights into molecular aspects of fat tail selection. These results may provide a strong foundation for studying the regulation of fat deposition in sheep and do offer hope that the causal mutations and the mode of inheritance of this trait will soon be discovered by further investigation.

**Abstract:**

The fat tail is a phenotype that divides indigenous Iranian sheep genetic resources into two major groups. The objective of the present study is to refine the map location of candidate regions associated with fat deposition, obtained via two separate whole genome scans contrasting thin and fat tail breeds, and to determine the nature of the selection occurring in these regions using a hitchhiking approach. Zel (thin tail) and Lori-Bakhtiari (fat tail) breed samples that had previously been run on the Illumina Ovine 50 k BeadChip, were genotyped with a denser set of SNPs in the three candidate regions using a Sequenom Mass ARRAY platform. Statistical tests were then performed using different and complementary methods based on either site frequency (F_ST_ and Median homozygosity) or haplotype (iHS and XP-EHH). The results from candidate regions on chromosome 5 and X revealed clear evidence of selection with the derived haplotypes that was consistent with selection to near fixation for the haplotypes affecting fat tail size in the fat tail breed. An analysis of the candidate region on chromosome 7 indicated that selection differentiated the beneficial alleles between breeds and homozygosity has increased in the thin tail breed which also had the ancestral haplotype. These results enabled us to confirm the signature of selection in these regions and refine the critical intervals from 113 kb, 201 kb, and 2831 kb to 28 kb, 142 kb, and 1006 kb on chromosome 5, 7, and X respectively. These regions contain several genes associated with fat metabolism or developmental processes consisting of *TCF7* and *PPP2CA* (OAR5), *PTGDR* and *NID2* (OAR7), *AR*, *EBP*, *CACNA1F*, *HSD17B10,*
*SLC35A2*, *BMP15*, *WDR13*, and *RBM3* (OAR X), and each of which could potentially be the actual target of selection. The study of core haplotypes alleles in our regions of interest also supported the hypothesis that the first domesticated sheep were thin tailed, and that fat tail animals were developed later. Overall, our results provide a comprehensive assessment of how and where selection has affected the patterns of variation in candidate regions associated with fat deposition in thin and fat tail sheep breeds.

## 1. Introduction

Identifying the regions of the genome that have experienced substantial selective pressure can provide a powerful tool into the location of functionally important polymorphisms and can help prioritize targets for association mapping [1,2]. Fat tailed breeds comprise approximately 25% of the world sheep population [3] and are grazed in a wide range of countries. The major biological role of the fat tail is to serve as an energy store against periodic food scarcity, such as in drought and winter [4]. It has also been used as a source of fat (or ghee) for human consumption for millennia [5]. Historically, the climatic conditions in these areas, as well as the nutritional and religious food requirements of the people, encouraged sheep producers to select for higher fat tail weight [6]. Although fat-tailed sheep breeds are still preferred under local pastoral, desert, or semi-desert conditions in several societies [7], most of the advantages of a large fat tail have recently reduced in importance due to the improved forage availability and decreased price for the product, especially in genetic improvement programs [4,8].

Various genomic-based studies have been conducted to understand the genetic basis and genomic architecture of sheep tails and to find the specific causal genomic variant(s) contributing to sheep tail pattern making [3,9,10,11,12,13,14,15]. A brief overview on genomic outcomes, including proposed potential genes generated from investigations on the sheep tail phenotype has been reported by Kalds et al. [7]. Additionally, transcriptomic analyses were performed, providing sets of potential genes that contribute to the formation and the biological emergence of sheep tails [16,17,18,19]. Taken together, these research efforts have revealed several high-ranking candidate genes with no current consistency or solid opinion about their variant causalities and expression nature. Therefore, to date the genetic mechanism underlying this particular trait has not been fully elucidated and finding the regions associated with fat deposition and the nature of the selection occurring in these locations are two of the most important and challenging areas of research in the countries grazing these breeds.

Genome scan approaches, such as quantitative trait locus (QTL) and hitchhiking mapping, provide the opportunity for investigating the genetic regions associated with different traits. QTL mapping studies have been applied to examine the genetic basis of various economically important traits, e.g., [20,21]. However, QTL mapping basically relies on detecting correlations between genetic markers and phenotypic traits in a segregating population [22]. Also, many traits underlying adaptive divergence are not always easily detectable at the phenotypic level and not well suited to QTL mapping. In this case, one approach for discovering the potential genetic basis of different traits is to use hitchhiking mapping [23,24]. Hitchhiking mapping is a population genetics approach use to identify genomic regions influenced by selection [25,26]. A great advantage of this approach is that it can be performed using molecular markers alone [27,28]. This approach is especially suitable for the traits such as sheep tail pattern where all sheep breeds can be easily classified in different classes (e.g., thin and fat tailed sheep groups) and then, offer ideal materials for comparative analysis of their genetic basis.

Hitchhiking mapping starts with a genome scan using an approximately uniformly spaced set of molecular markers followed by a fine scale analysis of the candidate regions [23]. For example, a marker-based analysis of chromosome 1 in rats from warfarin-resistant populations revealed a 0.5 centimorgan (cM) region that was the likely locus for this trait [29]. This information was used later to identify the warfarin resistance gene through more traditional candidate gene approaches and association mapping techniques [30].

Genomic scans for finding candidate regions associated with fat deposition in thin and fat tailed breeds have been described previously [3]. In brief, two independent experiments, including Iranian and ovine HapMap genotyping data contrasting thin and fat tailed breeds were analyzed, and using different statistics, especially F_ST_, three regions on chromosomes 5 (between 47,149–47,263 kb), 7 (between 46,642–46,843 kb) and X (between 58,621–61,452 kb) were confirmed in both data sets. Interestingly, these regions have also been supported in recently published studies using two different ways consisting of different breeds of thin and fat tailed sheep breeds through a genome-wide analysis of transcriptomic data [17,31] and selective sweeps [32,33], and also using same the breeds but different animals [34].

Unlike F_ST_ that has been used in the previous study [3], tests based on linkage disequilibrium (LD) like iHS [35] and XP-EHH [36], are multi-marker tests. These tests are commonly used for human SNP data, where there are now millions of SNPs available (approximately one marker per 1 kb), and that which depend on SNP spacing and frequency [37]. Densely spaced SNPs give greater power when using statistical tests that rely on LD, as signals of selection are less likely to be lost [38]. The Illumina Ovine SNP50k BeadChip, while providing uniform genome wide coverage, has a marker about every 56 kb [3]. Fine mapping, where more SNPs are genotyped in an area of interest, improves the ability to localize causal variants.

The allele frequency spectrum (F_ST_ and median homozygosity) and haplotype based (iHS and XP-EHH) statistics used in the current study have been shown by previous power analyses to be largely complementary [39]. F_ST_ and median homozygosity are single-marker tests and detect highly differentiated alleles between populations, where positive selection in one area causes larger frequency differences compared to neutrally evolving alleles. iHS, a measure of within breed evidence for selection, has a suitable power to detect partial selective sweeps. However, a disadvantage of this approach is that it loses power when the beneficial allele is close to fixation, because fixation will eliminate variation at and near the selected site. In contrast, XP-EHH can detect selected alleles that have risen to near fixation in one but not another population. Grossman et al. [40] showed that the F_ST_ and XP-EHH signals peaked more narrowly around the causal variant, making them useful for spatial localization, while iHS is better to distinguish causal variants and contributed little to spatial resolution. These tests were relatively uncorrelated in neutral regions, and only weakly correlated for neutral variants within selected regions [40].

In this study, three regions recognized as candidate regions associated with fat deposition in the tail were chosen for further analysis. This was achieved through fine-mapping with additional SNPs, using Sequenom technology. The aim of this study was to confirm the signature of selection in the candidate loci, narrow down the chromosomal region showing the selective imprint and to determine the nature of the selection process occurred in these regions. Our analyses focused on two site frequency (F_ST_ and Median homozygosity) and haplotype based (iHS and XP-EHH) statistics, and using this variety of selection sweep tests we were able to obtain some novel results associated with fat deposition in these breeds.

## 2. Materials and Methods

### 2.1. Animal Sampling and Genotyping

Two independent data sets consisting of Zel-Lori Bakhtiari and Ovine HapMap samples were previously used for a genome-wide scan analysis of selective sweeps in thin and fat tail breeds [3]. Because the latter project did not formally phenotype the individuals concerned and we did not have access to their DNA samples, in this research further analysis was performed for the Zel-Lori Bakhtiari data. This data set consisted of 45 samples each of the thin (Zel) and fat tailed (Lori Bakhtiari) breeds (37 Females and 8 Males per each breed). The animals were collected to be as unrelated as possible. Three regions (Table 1), identified as being under selection based on F_ST_ and median homozygosity results from the first genome scan, were chosen for further analysis.

Once the regions of interest were defined, all known SNPs in each region were examined for suitability for genotyping. These included SNPs discovered on both the Solexa and 454 platforms (http://www.sheephapmap.org/genseq.php, accessed on 31 May 2017). A total of 156, 89, and 78 putative SNPs flanked by 150 bp on each side (301 bp total) were submitted to Sequenom’s primer design software (Mass ARRAY Assay Design 3.1) for the regions on chromosomes 5, 7, and X, respectively, and primers and probes were designed in multiplex format. Of these SNPs, assays were successfully designed for 140 (90%), 75 (84%), and 66 (84%) of all SNPs initially selected for study, and were grouped into 6, 3, and 4 multiplex assays for the three regions respectively. The remaining SNPs failed primer design, primarily due to high repeated content. The majority of these multiplex assays were located in the first and second plex which contained 71, 62, and 57 SNPs for chromosomes 5, 7, and X respectively. These two SNP assays were used for genotyping. Selected SNPs, their locations, PCR primers, unextended primer, and its extension masses, are represented in Appendix A. All SNPs were genotyped by use of the mass-spectrometry based iPlex Mass Array platform provided by Sequenom (Sequenom, San Diego, CA, USA. http://www.sequenom.com/, accessed on 31 May 2017). Experiments were conducted following the Sequenom iPLEX Assay application note [41].

### 2.2. Quality Control Filters

All samples with more than 30% missing data and subsequently all loci with more than 15% missing data were excluded. These rejection thresholds were chosen by plotting numbers of animals or loci against percent missing data and the cutoff point was determined as the curve inflection point, where the rate of change in the number of excluded loci, became linear with every increased percent of missing data [42]. For the remaining SNPs, those with a minor allele frequency (MAF) of less than 10% over all samples, and an outlier departure from Hardy—Weinberg equilibrium over all animals of a breed (*p*-value < 0.0002), were excluded [3,43]. We combined the SNP genotyping results from both the Ovine SNP50k BeadChip (in the regions of interest) and the Sequenom Mass ARRAY platform for final analysis. Across the three regions, 25 SNPs were genotyped on both Sequenom and Illumina platforms with a mean allelic concordance rate of 96% (range 92–99%). For the few SNPs with different genotypes, the Ovine SNP50k Bead Chip results were used due to their greater accuracy.

### 2.3. Estimates of F_ST_ and Median Homozygosity

To determine the pattern of positive selection, the basic form of Wright’s fixation index (F_ST_) was calculated as described by MacEachern et al. [44]. The value of F_ST_ can theoretically range from zero (showing no differentiation) to one (indicating complete differentiation, i.e., populations are fixed for different alleles). For each set of five adjacent SNPs, the average of F_ST_ values was calculated and termed windowed F_ST_. This is an approximate method of looking for regions where selection is apparent over multiple markers, rather than one-off high values. A window of five markers was chosen as it appeared to provide the better signal compare to other arbitrary window sizes [3]. The windowed F_ST_ values were then plotted against genome location. 

All analyses presented in this work were also performed using Weir and Cockerham [45] and Hudson [46] methods. The results were almost identical for all F_ST_ estimators (r ≥ 99%), so that we have only presented F_ST_ results based on Wright’s estimator in the present study for an easier comparison to previous reported results [3]. All scripts for estimating Wright (F_ST_), Weir and Cockerham, and Hudson’s fixation indices were written and performed in R v 4.0.2.

The selection for a new beneficial mutation increases the level of homozygosity in the selected allele and neighboring regions due to the hitchhiking effect. Therefore, one method of looking for the region where selection has taken place is to compare median runs of homozygosity between breeds. To do this, the median run of homozygosity for each SNP was calculated following Moradi et al. [3] in each animal. The length of a run of homozygosity, that is the number of consecutive homozygous SNPs including the one being considered, was calculated (this would be zero if the SNP being considered was heterozygous). For each marker the median length, over the breed, of the run of homozygosity was calculated and plotted against genomic position in the candidate regions (25 SNPs on each side).

It should be noted that since the historical effective population sizes of males and females are not the same for sexual and autosomal chromosomes [47], all analyses presented in this paper for chromosome X were performed by using only females (37 animals per each breed), although the analysis for both sexes produced similar results.

### 2.4. Determining of Ancestral Alleles

To calculate iHS and XP-EHH, the ancestral allele state of each SNP must be specified. Ancestral alleles for the ovine chip SNP were obtained from international sheep hapmap project (ISGC), and then to obtain the ancestral alleles for the additional Sequenom SNPs, 301 base pairs of sequence (1 bp of the alleles plus 150 bases either side of the SNP) were aligned against *Bos taurus* (cattle), *Sus scrofa* (pig), *Equus caballus* (horse), *Canis familiaris* (dog) and *Homo sapiens* (human) genomes using BLAST (http://blast.ncbi.nlm.nih.gov/Blast, accessed on 31 May 2017). A cross-species megaBLAST of Sequenom^®^ primers was used to discover ancestral alleles for the remaining SNPs [37]. The ancestral allele was taken as the base in the genome sequence at the resulting SNP position. For loci where only one SNP allele was represented in the other species, that allele was determined as ancestral. For other SNPs, as an additional tool in determining the ancestral allele, a phylogenetic tree of the five species was used [48]. This provided a crude tree, which could be used in decision making; for example, if all the animals had the same allele (C) apart from humans (T), then the ancestral was more likely to be C, as humans are more distantly related than the other species. In this research, we were able to determine the ancestral status of 36, 28, and 36 from the 41, 30, and 36 SNPs which passed quality control in the regions on chromosome 5, 7, and X respectively.

### 2.5. Reconstruction of Haplotypes

A pair of haplotypes was reconstructed for each animal in the sample using fastPHASE version 1.2.3 [49]. This software implements an Expectation—Maximization strategy for estimating missing genotypes and for reconstructing haplotypes from unphased SNP genotypes data of unrelated individuals [49].

### 2.6. Calculation of Integrated Haplotype Score (iHS) and Cross-Population EHH (XP-EHH)

iHS was calculated as in Voight et al. [35] and XP-EHH as in Sabeti et al. [36]. These statistical tests were calculated using the rehh package [50] in R v4.0.2 and the candidate genomic regions under selection were obtained.

Briefly and following Voight et al. [35], the iHS was computed for every SNP with ancestral state information and MAF above 10% (Appendix A). This test is based on the extended haplotype homozygosity (EHH) statistic [39], which measures the decay of identity, as a function of distance, of haplotypes that carry a specified core allele at one end. The integral of the observed decay of EHH with distance from the core allele is calculated until EHH reaches 0.05. This integrated EHH (iHH) (summed over both directions away from the core SNP) is denoted as iHH_A_ or iHH_D_, depending on whether it is computed for the ancestral or derived core allele. The unstandardized iHS (uiHS) was then calculated as ln (iHH_A_/iHH_D_). The uiHS is thus adjusted so that the final statistic has a mean of 0 and a variance of 1, regardless of allele frequency at the core SNP. To do this, the results for each breed were split into 18 equally sized allele frequency bins, from which a mean and standard deviation were calculated. These were used in the following equation [35]:$$\mathrm{iHS}=\frac{\mathrm{uiHS}-E_{p}\left[\mathrm{uiHS}\right]}{{\mathrm{SD}}_{p}\left[\mathrm{uiHS}\right]}$$
where the expectation and standard deviation are calculated using SNPs from the same bin (*p*). Due to the different demographic histories of the X chromosome and the autosomes (e.g., due to smaller effective population size), we normalized the iHS scores of this chromosome separately from those of the other chromosomes. Results of iHS are presented here as |iHS| for a window of 10 SNPs and then plotted against genome location. We chose a window of 10 SNPs because of the longer extent of LD and SNP spacing in sheep compared to humans, in which the window length used is commonly around 40 SNPs [35]. Large positive and negative values of iHS indicate unusually long haplotypes carrying the ancestral and derived allele, respectively; by taking the absolute iHS value, interesting variants will be shown by large positive values.

XP-EHH compares haplotypes between populations to control for local variation in recombination rates [36]. Briefly, XP-EHH is defined relative to a given SNP *i* in two populations, A and B. In each population, the expected haplotype homozygosity (EHH) [39] was integrated with respect to genetic distance in both directions from *i*. The log of the ratio of these integrals, ln(*I_A_/I_B_*), is the abnormal XP-EHH (for more details see [36]). XP-EHH must also be normalized for genome-wide differences in haplotype length between populations, so that there is a mean of zero and unit variance. This was done by subtracting the mean and dividing by the standard deviation of all scores. In this study, we defined the XP-EHH test with respect to thin and fat tail breeds and an unusually positive value suggests selection in fat tailed population and a negative value selection in thin tailed.

### 2.7. Core SNP Alleles and Haplotype Frequencies in Candidate Regions

After the aforementioned filtering process and reconstruction of haplotypes for candidate regions using PHASE 2.1 [51], the haplotypes were fed into SWEEP v.1.1 [39] to detect core regions based on the EHH statistic in candidate regions, which is fully described by Sabeti et al. [39]. PHASE was chosen here over fastPHASE as haplotype estimates are slightly more accurate using PHASE (http://stephenslab.uchicago.edu/software.html, accessed on 1 February 2021), although substantially slower to compute. For selection of core regions and study of haplotype frequencies in selected area of our interested regions, the results of F_ST_, iHS and XP-EHH tests were also used as additional information. Finally, the pattern of haplotype blocks in the regions of interest were constructed and the decay of LD (pairwise r^2^) was visualized using Haploview [52].

### 2.8. Study of Identified Genes in Candidate Regions

Genes located in the genomic regions significantly differentiated between sheep breeds were acquired by the use of the data mining tool Biomart (http://asia.ensembl.org/biomart/martview, accessed on 1 February 2021), with the reference assembly of the *O. aries* genome OAR v3.1 [53]. The regions of interest in *O. aries* were also compared to the corresponding areas in *B. taurus* as its genome is better annotated. Regions chosen for fine mapping onto OARv3.1 and their orthologous coordinates in *B. taurus* (ARS-UCD1.2, Bostau9) is shown in Appendix A. It should be noted that, due to the easier comparison of coordinates obtained by the current study with the previous article [3], the coordinates have been presented for different statistics in this study, based on OAR v1.0. However, the coordinate of candidate regions associated with fat deposition reported in Moradi et al. [3] and the fine mapped results in this study have been also shown based on different OAR versions in Appendix A. Nearby genes within a flanking distance of 500 kbs from each region were acquired. This distance has been selected as previously considered by Moradi et al. [3] for sheep and by Do et al. [54] for Holstein dairy cattle. To determine the biological functions of each gene, Biological Process (BP) and Molecular Functions (MF) of all identified genes in candidate regions were studied using DAVID annotation [55]. Furthermore, a comprehensive literature review was conducted to verify whether these genes have some relevance with fat deposition or developmental process in sheep or other mammals.

## 3. Results

### 3.1. SNP Genotyping and Data Mining

A total of 32, 26, and 29 SNPs passed the filtering criteria in our regions of interest on chromosome 5, 7 and X respectively (Table 2). Most of these SNPs that used for further analysis were not available on the Illumina Ovine SNP50k BeadChip due to the lower expected minor allele frequency (MAF) and reliability of the SNPs, discovered on both the Solexa and 454 platforms (http://www.sheephapmap.org/genseq.php, accessed on 31 May 2017).

Table 2 presents a descriptive summary of the characteristics of the SNPs used in the final analysis. The distribution of SNPs varied among the regions, especially regarding to those used in statistical tests (Appendix A); however, the average SNP intervals were relatively consistent and the overall average distance between adjacent SNPs was about 23 kb, 17 kb and 82 kb for candidate regions on chromosome 5, 7 and X respectively, compared to about 60 kb and 115 kb for autosomes and chromosome X on the Illumina Ovine SNP50k BeadChip respectively.

### 3.2. Distribution of F_ST_ and Median Homozygosity

To identify loci that have been targets of selection across thin and fat tailed breeds, the windowed F_ST_ was plotted against genomic location for candidate regions (Figure 1a). Variants with unusually large F_ST_ values are typically interpreted as being the targets of local selective pressures due to the hitch-hiking effect [25].

The average of differentiation between Zel (thin tail) and Lori Bakhtiari (fat tail) for the whole genome was 0.024 (SD = 0.036), while this parameter was 0.093 (SD = 0.136), 0.172 (SD = 0.184) and 0.279 (SD = 0.217) at the areas of interest on chromosome 5, 7 and X respectively. As shown in Figure 1, in these regions we found evidence of selection across relatively short distances with windowed F_ST_ values > 0.30 on chromosomes 5 (between 47,149,400–47,245,841 bp), 7 (between 46,587,943–46,843,356 bp) and values > 0.40 on chromosome X (between 59,257,971–59,984,949 bp). The score of 0.30 is in the 99.9 percentile of autosomal SNPs (*n* = 44,558) and a score of 0.40 is in the 99.0 percentile of chromosome X SNPs (*n* = 1126).

When there is a selection for a causal mutation in one breed and not the other, the breed under selection shows high homozygosity in a genomic interval while the other does not [39]. To further test this hypothesis, median homozygosity was calculated and plotted against genomic position for candidate regions (Figure 1b). The results indicate that homozygosity increased over the areas of interest on chromosome 5 and X for the fat tailed and at the candidate region on chromosome 7 for the thin tailed breed. The largest differences of median homozygosity were for chromosome X and homozygosity was present for a longer distance as well, whereas these parameters are less on Chromosome 5.

### 3.3. Calculation of Integrated Haplotype Score (iHS)

iHS detects signatures of strong selection in favor of alleles that have not yet reached fixation [35]. The results revealed that regions on chromosomes 5 and X had no iHS peak (Appendix A). This observation may suggest that selected alleles have already been fixed in these locations. However, analysis of the region on chromosome 7 displayed an obvious peak in both thin and fat tail breeds (Figure 2). For most locations of the region the values are higher in the thin tail breed, however, there is also some even stronger indication of selection in fat tail breed especially at the end of the region.

### 3.4. Cross-Population EHH (XP-EHH)

To identify selective sweeps in which the selected allele has approached or achieved fixation in a subpopulation, but remains polymorphic in the population as a whole, the standardized cross population extent of haplotype homozygosity scores (XP-EHH) were plotted against genomic locations (Figure 3). The results revealed clear peaks in all of our regions of interest, suggesting that selected alleles approached fixation or have risen to near fixation in favor of fat tail breed on chromosome 5 and X while on chromosome 7 the frequency of alleles have risen close to fixation in thin tail breed. These results are based on linkage disequilibrium around given SNP and are in agreement with the results of median homozygosity plots.

Sabeti et al. [36] considered a region as a candidate for selection in the human HapMap Phase 2 dataset when the two population XP-EHH was above 4.34. This score is in the 99.9 percentile for thin versus fat tail breeds. As shown in Figure 3, we found evidence of selection with |XP-EHH| value > 4.34 on chromosomes 5 (47,141,229–47,171,110 bp), 7 (46,604,500–46,642,359 bp) and X (59,187,456–60,264,325 bp). These regions are almost identical to the positions with highest F_ST_.

### 3.5. Study of Core Haplotypes and Their Ancestral Status in Candidate Regions

To evaluate the haplotype frequencies and the status of selected alleles (derived or ancestral), the core SNP alleles, and their haplotype frequencies were investigated in the selected regions (Table 3).

For chromosome 5 (Table 3), we defined a core region of 26 k where both F_ST_ and XP-EHH statistics were at their highest. There are 5 genotyped SNPs in this region. The SNPs defined 14 core haplotypes (denoted haplotype 1 to 14) in thin tailed but only 6 core haplotypes in fat tailed sheep. As shown in Table 3, there is a common haplotype (haplotype 1) with a frequency of 90% in fat tailed sheep whereas its frequency for thin tailed sheep is 15%. In contrast, the common haplotype in thin tail breed is haplotype 8 with a frequency of 31%, while it is almost absent (2%) in the fat tail breed. The interesting result in this region is that all the SNPs in the common haplotype for the fat tail breed are derived SNPs whereas all SNPs in the common haplotype for the thin tail breed are ancestral.

For chromosome 7 (Table 3) we defined a core region of 37 k in the selected area with 4 genotyped SNPs, where all statistics showed strong evidence of selection. There were only 4 core haplotypes (denoted haplotype 1 to 4) in thin tailed and 6 haplotypes in fat tailed sheep. The common haplotypes of fat and thin tail breeds in this region were haplotype 5 and 1 with frequency of 31% (2% in thin tailed) and 80% (6% in fat tailed), respectively. Once again, while the common haplotype in the fat tail breed had all derived alleles, the common haplotype in the thin tail breed were almost all ancestral alleles (3 out of 4 SNPs). However, it is notable that in this region the haplotypes appear more polymorphic in the fat tail breed and the frequency of the common haplotype of the thin tail breed is not as high as for the other candidate regions.

Subsequently, this approach was performed in the candidate region on chromosome X. We defined a core region of 242 k corresponding to 6 genotyped SNPs. The longer length of the core region, in this case is due to the wider SNP spacing on chromosome X. The genotyped SNPs defined seven core haplotypes in the thin tail breed and only two haplotypes in the fat tail breed (Table 3). The common haplotype in the fat tail breed was haplotype 1 with a frequency of 89%, whereas its frequency in thin tailed was 12%. In this region the common haplotype in the thin tail breed (54%) was absent in the fat tail breed. The results in this region were similar to the previous regions in that the derived alleles were more prevalent in the common haplotype of fat tailed (4 out of 6 core SNPs) with the same result for ancestral alleles in the common haplotype of the thin tail breed. Together, these results could suggest that ancestral alleles have been under selection in the thin tail breed while the contrary happened for the fat tail breed where derived alleles have undergone selection.

### 3.6. Study of the Identified Genes Associated with Fat Metabolisms in Candidate Regions

The regions of interest were investigated to determine if any genes related to fat deposition could be identified in sheep or their corresponding areas of the cow genome. The genes obtained in *O. aries* or by orthology with *B. taurus* and their functions are presented in Table 4.

There is also the hypothesis that some genes selected for, in fat tail sheep breeds are likely to be also associated with developmental defects or ectopic expression of organs [56,57]. Our results revealed that these regions contain many genes, having some known biological functions associated with the developmental process in *O. aries* (Table 5) and their orthologous coordinates in *B. taurus* (Appendix A).

## 4. Discussion

An increasing number of studies have been conducted to detect signals of recent positive selection on a genome-wide scale in different domestic animals [58,59,60]; however, there are relatively few genomic regions identified that have been subject to selection for a specific mutation underlying evolutionary shifts in a trait. In this paper, in order to fine map and get more insight into the genomic basis of fat deposition in thin and fat tail breeds, we have investigated three candidate regions using this approach. These candidate regions were analyzed using a variety of statistics to clarify the signals of selection observed in these regions. Our analyses focused on two allele frequency spectrum (F_ST_ and median homozygosity) and haplotype based (|iHS| and XP-EHH) statistics. These tests were chosen because previous power analyses suggested that these are largely complementary [39].

The study of the candidate regions on chromosome 5 and X revealed obvious evidence of the selection using an F_ST_, median homozygosity, and XP-EHH test in a relatively narrow region; while an examination of these regions identified no particular |iHS| peak. With a hypothesis that historically different selection pressures operated in thin and fat tail breeds and somehow selection acted on a variant that was advantageous only in one breed, these results suggest that selection in these regions occurred for mutations affecting fat tail size as the beneficial mutations have risen to near fixation in fat tailed breeds. This suggestion is also supported by the core haplotype frequencies observed in candidate regions on these chromosomes as the common core haplotypes in fat tail breeds were near fixation (Table 3).

Analysis of the region on chromosome 7 indicated strong evidence of selection using all selective sweep statistics. The results of F_ST_ and median homozygosity suggested that the selection differentiated the beneficial alleles between breeds and that homozygosity has been increased in favor of thin tailed in this region. However, |iHS| and XP-EHH revealed additional information. |iHS| provided evidence of partial selection in both breeds, while the XP-EHH results showed that the selected alleles have approached fixation in the thin tail breed.

As discussed earlier, the power of the |iHS| statistics to detect selective sweeps is greatest at a moderate allele frequency (~40–60%), while XP-EHH test is more powerful for detecting selective sweeps close to fixation (>80%) [35,61]. However, both of these methods do have power outside their optimal ranges. Sabeti et al. [36] demonstrated that the iHS statistic could detect signals over the range of 20–80% and XP-EHH do have power between 60–100%. Overall, these results suggest that while the frequency of selected alleles has been raised to fixation in thin tail breed, its frequency should be ~60–80% to be picked up by both methods. Simultaneously, the favorable allele should be increased to mid-range frequency in the fat tail breed. The results of core haplotype frequencies in this region (Table 3) are in consistent to this point as the common haplotype in thin tail breed has a frequency of 80%, whereas all haplotypes observed in this region appears polymorphic in the fat tail breed. One inference is that there has been an ongoing infusion of fat tailed haplotypes into this breed, but also selection for the thin-tailed phenotype.

To further test this hypothesis, we constructed the pattern of haplotype blocks in this region and the decay of LD (pairwise r^2^) was visualized using Haploview [52]. The effect of a selective sweep on patterns of variation is expected to decline with time (due to recombination) and if a selective sweep is still ongoing in a subpopulation, the hitchhiking haplotype is expected to be rather long [39,57]. Our results (Appendix A) revealed that although there were haplotype blocks in both breeds, they extended for longer distances in the fat tailed compared to the thin tail breed. This suggests that as the prevalence of selected alleles increased in thin tail breed, the LD around variants decayed due to recombination, while the selection in the fat tail breed is younger (longer haplotype blocks). These results confirm our observations for evidence of selection in this region with both |iHS| and XP-EHH tests.

The earliest known depiction of a fat tail sheep is on an Uruk III stone vessel about 5000 years before present, approximately 4000 years after initial domestication [62]. Given that fat tailed breeds are now prevalent in the Fertile Crescent, where sheep were originally domesticated, while thin tailed sheep breeds are predominant in peripheral areas and that the wild ancestor of sheep is thin tail, it has been assumed that the first domesticated sheep were thin tailed and fat tail was developed later [62,63]. We have investigated this hypothesis through the classification of selected alleles in the core haplotypes of our regions of interest as ancestral or derived. Our results provide the preliminary molecular evidence to confirm this assumption since, we observed that in almost all cases, derived alleles have been under selection pressure in the fat tail breed, and this is consistent with the selection of a new mutation in these breeds (Table 3).

Population demographic history can also cause similar patterns on DNA sequence variation and could be a source of error in making inferences on genomic targets of selection. This caveat can be avoided by screening a large number of markers (as has previously been performed in this research) spaced across whole genome [3], as selection will result in regional patterns compared to the genome-wide effects of population history and demographic events [23,64]. Similarly conducting this type of fine-scale analysis at the candidate genomic regions using a dense set of markers and multiple statistical tests, reduces the chance that a signature of selection will be a false positive if it is detected in more than one marker locus and statistical test [40].

The candidate regions of interest have previously been studied using OAR v1.0 in *O. aries* and their corresponding area of *B. taurus* and no particular candidate genes associated with fat deposition were identified [3]. In this study, investigation of these regions using the newly available sheep genome OAR v3.1 [53], defined some genes associated with fat metabolism in *O. aries* or their orthologous areas of *B. taurus* (Table 4). Protein phosphatase 2 (formerly 2A), catalytic subunit, alpha isoform (*PPP2CA*) has a variety of roles in different biological process such as cellular lipid metabolic, membrane lipid metabolic and sphingolipid metabolic process, while hydroxysteroid (17-beta) dehydrogenase 10 (*HSD17B10*) and emopamil binding protein (*EBP*) genes play some roles in lipid metabolic process and androgen receptor (*Ar*) and synaptophysin (*Syp*) genes get participate in lipid binding [65]. Interestingly, most of the genes (or their gene families) identified here, have been recently reported as candidate genes associated with lipid metabolisms using various molecular techniques in sheep. Yuan et al. [66] implemented differential expression analysis using RNA-seq technology in longissimus dorsi muscle tissue (MUT), perirenal adipose tissue (PAT) and tail adipose tissue (TAT) of different Chines short and fat tailed sheep breeds and revealed that *PPP1CA* is highly expressed in TAT. Also, *PPP1CA* was identified as plausible genes associated with the fat-tailed or fat-rumped phenotype by comparing copy number variations (CNVs) with different tail types [67]. Moreover, protein phosphatase 1, catalytic subunit, gamma isozyme (*PPP1CC*) have been under selection signature in Chinese thin and fat-tailed sheep breed and reported to be associated with tail type [68]. *HSD17B12* has also been reported to be associated with fat tail metabolism in thin and fat tailed sheep breeds using deep transcriptome analysis with RNA-Seq data [69]. It is reported that *HSD17B12* (act as elongates) are important genes for controlling the overall balance of fatty acid composition [69]. Therefore, it seems the gene families and isoforms of Protein phosphatase (PPP) and hydroxysteroid (HSD) have important roles in molecular regulations of fat deposition in tail.

To confirm the results of our study, the exon 1 of *PPP2CA* gene was amplified and its variation patterns were sequenced in an independent study on Zel and Lori-Bakhtiari sheep breeds [34]. Two patterns were identified and the results of sequencing showed that in Lori-Bakhtiari, Del/Del genotype resulted in heavier fat tail than T/T genotype (5.20 ± 0.21 kg vs. 3.28 ± 0.12 kg) (*p* < 0.05) while, in Zel, the effect of genotypes on carcass fat percentage and triglyceride was significant, so that the T/T genotype had more carcass fat percentage comparing to Del/Del genotypes (*p* < 0.05). Overall, it seems as the annotation of the ovine genome becomes more complete, all genes located in the candidate regions will be identified and promising targets can then be verified by further experimentation.

A result which is irrelevant to the inheritance of the trait, but provides an insight into a possible mechanism of fat deposition in this organ, are the results of Gokdal et al. [56] who examined the effects of docking in fat tail breeds. The carcasses of the docked group contained more kidney, pelvic and internal fat than the intact lambs as well as a higher percentage of subcutaneous and intramuscular fat. The weights of the different carcass cut of the docked lambs were also heavier than those of the intact group. However, there was little change in overall carcass composition, suggesting that the genes affecting the fat tail phenotype are associated with the localization of fat stores to a regional depot rather than control of the overall level of fat deposition. This observation also may provide support to the suggestion that some genes selected for in fat tail sheep breeds in these regions are likely to be also associated with developmental defects or ectopic expression of organs. Our results revealed that these regions contain many genes, having some known biological functions associated with developmental process (Table 5 and Appendix A). Several earlier studies provide evidence for this issue. For example, the transcription factor (*TCF*) genes have been among highly differentially expressed genes in perirenal adipose tissue (PAT) and identified as being the most likely to account for the fat-tailed phenotype of sheep [66]. *TCF7* is involved in the Wnt/β-catenin signaling pathway, and this pathway plays a critical role in regulating sheep [69] and porcine [70] adipogenesis genes expression. Zhu et al. [71] studied copy number variations (CNVs) and selection signatures on the X chromosome of Chinese indigenous sheep with different tail types and revealed that the regions harboring CNVs and selective sweeps in different sheep breeds overlapped with calcium channel, voltage-dependent, L type, alpha 1F subunit (*CACNA1F*) gene that could be as associated with tail type in these breeds [43]. In addition, *HSD17B10* [69], *SLC35A2* [68], *AR* and *TIMP1* [31], have been identified as candidate genes that affect fat tail development. Moreover, it is important to note that our regions of interest overlapped with some genes, for example *BMP15*, *WDR13* and *RBM3* that belong to the gene families that their closely related genes consisting *BMP2* [9,32,68,72,73], *WDR92* [17,68] and *RBM11* [6] have recently reported to be associated with fat tail formation and adipose tissue gene expression in sheep.

Finally, fine mapping of candidate regions using different sweep statistical tests has enabled us to confirm the signature of selection in these chromosomal regions and better refine the critical regions from 113 kb (47,149,400–47,263,230) to 28 kb (47,146,931–47,175,489) on chromosome 5, from 201 kb (46,642,359–46,843,356) to 142 kb (consisting to shorter intervals: 46,587,943–46,642,359 and 46,765,080–46,852,870) on chromosome 7 and from 2831 kb (58,621,412–61,452,816) to 1006 kb (59,257,971–60,264,325) on chromosome X. These regions were refined considering all statistical test results (Appendix A). Acquiring of the genes located within the regions of interest after fine mapping revealed some genes consisting *TCF7* on chromosome 5, *PTGDR* and *NID2* on chromosome 7 and finally *AR* on chromosome X that have a multiple effect on lipid metabolisms, macromolecule metabolic process, organ/gland development or associated with ectopic expression of organs simultaneously.

Recently published study on the origin of European sheep as revealed by the diversity of the Balkan breeds and by optimizing population-genetic analysis tools [74] using a variety of sheep breed samples from Southwest-Asian, Mediterranean, Central-European and North-European showed that the thin-tailed Zel sheep is found to be in the same genetic cluster as the fat-tailed Iranian sheep, whereas the fat-tailed Italian Laticauda is related to other breeds in central Italy. This may imply that the tail phenotype is encoded by a limited number of genes. By combining information of the present study, previously reported and annotated biological functional genes, we suggest *PPP2CA* and *TCF7 (OAR5)*, *PTGDR* and *NID2* (OAR7), *AR*, *EBP*, *CACNA1F*, *HSD17B10*, *SLC35A2*, *BMP15*, *WDR13* and *RBM3* (OAR X) as the most promising candidate genes for type of tail traits.

It is obvious that understanding the mechanisms that underlie fat tail inheritance in sheep is difficult to verify solely by selective sweep profiling. While this study has now refined three regions located on chromosome 5, 7 and X, associated with fat tail sheep breeds, it is likely that other genomic regions may also be involved. Some other studies have identified a region on Chromosome 15 for example, close to *PDGFD* as associated with the fat tail phenotype [12,13]. Definitive studies of the actions of these regions will require larger-scale designs in segregating populations where trait measurements are recorded. Also, these regions may still be too large to efficiently implement technologies such as marker assisted selection or positional cloning. More detailed and larger scale experiments from these and other thin and fat tailed breeds may allow us to refine the location of the causal mutations. Likewise, it does not exclude contemporaneous selection for other traits, and any regions identified still need to be tested for a functional genetic relationship via trait measurement in contrasting genotypes or phenotypes. Specifically, future studies should be conducted in reciprocal F_2_ crosses to provide independent and causal evidence and verify the mode of inheritance. If these areas are shown to have a significant effect, then further work sequencing either side of the regions can help the search for causal variants.

## 5. Conclusions

Our results provide a comprehensive assessment of how and where selection has affected the patterns of variation in candidate regions associated with fat deposition in thin and fat tail sheep breeds. These results enabled us to confirm the signature of selection in these regions, refine the critical intervals regions, and to identify the most promising candidate genes associated with fat deposition in thin and fat tail sheep. These results may provide a strong foundation for studying the regulation of fat deposition in sheep and do offer hope that the causal mutations and the mode of inheritance of this trait will soon be discovered by further experimentation.

## Figures and Tables

**Figure 1 animals-12-01423-f001:**
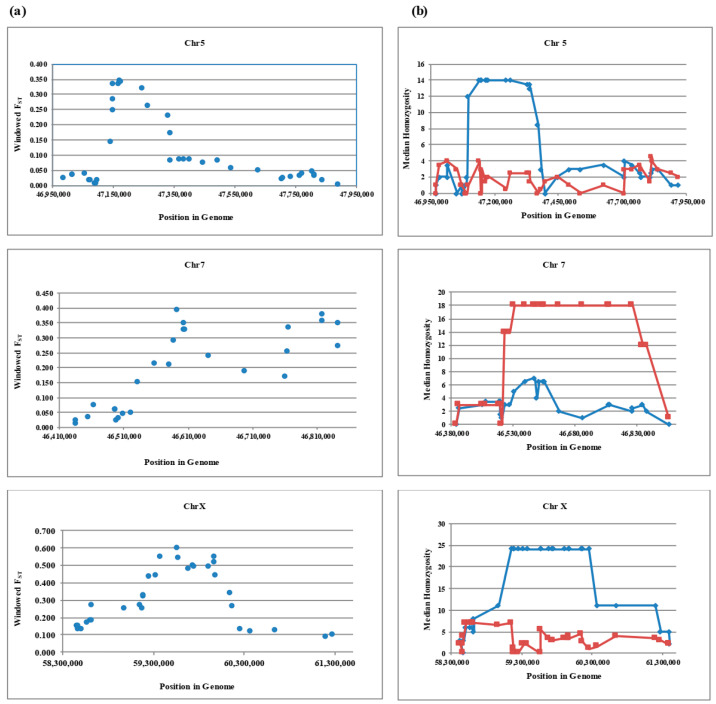
Plots of windowed F_ST_ (**a**) and run of median homozygosity (**b**) in relation to genomic position for thin and fat tail breeds in candidate regions: SNP positions in the genome (bp) are shown on the X-axis, and windowed F_ST_ or median homozygosity are plotted on the Y-axis. Fat and thin tailed breeds are shown by blue diamonds and red squares respectively on median homozygosity plot.

**Figure 2 animals-12-01423-f002:**
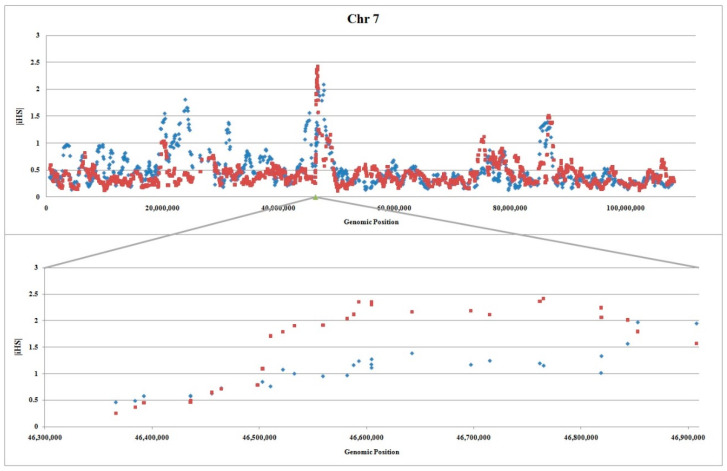
Plot of |iHS| in relation to the genomic position (bp) for thin and fat tail breeds on chromosome 7 (**Upper**) and our candidate region (**lower**): Fat and thin tail breeds are shown by blue diamonds and red squares respectively, and |iHS| statistic averaged over 10 SNPs. The information for calculating |iHS| on whole chromosome was obtained from Moradi et al. [3].

**Figure 3 animals-12-01423-f003:**
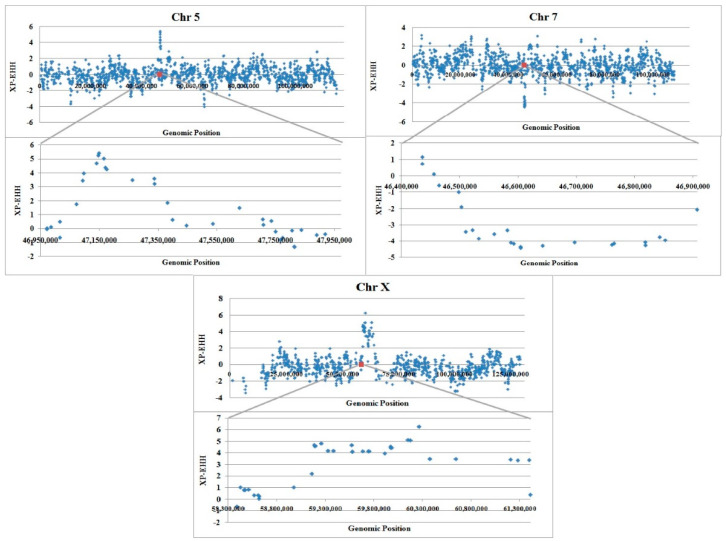
Plot of XP-EHH relation to genomic position (bp) for thin and fat tailed breeds on whole chromosome (**upper**) and our candidate region (**lower**) in different chromosomes: High positive values suggest selection in fat tailed population and negative values selection in thin tailed population. The genotyping information required for the presentation of the entire chromosomes were obtained from Moradi et al. [3].

**Table 1 animals-12-01423-t001:** Regions chosen for fine mapping in this study and the number of SNPs that were previously genotyped in these regions using the Ovine SNP50k BeadChip.

Chromosome	Region (bp)	Length (bp)	Ovine SNP50k BeadChip SNPs
5	46,971,979–47,919,440	947,461	15
7	46,392,398–46,852,870	460,472	10
X	58,424,602–61,409,447	2,984,845	20

**Table 2 animals-12-01423-t002:** Summary of SNP characteristics for different regions, genotyped using the Sequenom assay, before and after data cleaning and their combination with Ovine SNP50k BeadChip SNPs, for follow up analysis.

Chromosome	No. of SNPs Assayed with Sequenom	No. of SNPs That Passed Quality Control	Total Number of SNPs Used for Final Analysis	Average Distance between SNPs (kb)
5	71	32	41 (15 + 26) ^1^	23.69
7	62	26	30 (10 + 20)	17.77
X	57	29	36 (20 + 16)	82.91

^1^ SNPs included Ovine SNP50k Bead Chip + non repeated Sequenom SNPs.

**Table 3 animals-12-01423-t003:** Core SNP alleles and haplotype frequencies in candidate regions for fat tail (Lori) and thin tail (Zel) breeds: The haplotypes with higher frequency in fat and thin tailed breeds have been highlighted.

**Chromosome 5**								
		**Core SNP Alleles**					**Core Haplotype Frequencies**	
**SNP Variants**		**C/T**	**A/G**	**T/C**	**A/G**	**A/C**	**Fat Tail Breed**	**Thin Tail Breed**
Genomic position (bp)		47,149,354	47,149,400	47,165,900	47,171,110	47,175,489		
Ancestral allele		T	G	T	G	C		
Haplotype 1		C	A	C	A	A	**0.90**	0.15
Haplotype 2		- *	G	-	-	-	0.01	0.06
Haplotype 3		-	G	T	-	-	0.01	-
Haplotype 4		-	G	-	G	C	-	0.12
Haplotype 5		T	G	-	-	-	-	0.13
Haplotype 6		T	-	T	-	-	0.01	-
Haplotype 7		T	G	T	-	-	0.04	0.11
Haplotype 8		T	G	T	G	C	0.02	**0.31**
Other Haplotypes							-	0.14(6 Haplotypes)
**Chromosome 7**								
		**Core SNP Alleles**					**Core Haplotype Frequencies**	
**SNP Variants**		**C/T**		**A/C**	**T/C**	**A/C**	**Fat Tail Breed**	**Thin Tail Breed**
Genomic position (bp)		46,604,500		46,604,644	46,604,722	46,642,359		
Ancestral allele		C		C	C	A		
Haplotype 1		C		C	C	C	0.06	**0.80**
Haplotype 2		-		-	-	A	0.20	0.14
Haplotype 3		T		A	-	-	0.02	0.03
Haplotype 4		T		A	-	A	0.14	-
Haplotype 5		T		A	T	-	**0.31**	0.02
Haplotype 6		T		A	T	A	0.27	-
**Chromosome X**								
	**Core SNP Alleles**						**Core Haplotype Frequencies**	
**SNP Variants**	**A/G**	**C/T**	**T/C**	**T/A**	**G/C**	**G/A**	**Fat Tail Breed**	**Thin Tail Breed**
Position (bp)	59,742,181	59,750,338	59,912,586	59,971,891	59,971,909	59,984,949		
Ancestral allele	A	T	T	T	C	A		
Haplotype 1	G	C	T	T	G	G	**0.89**	0.12
Haplotype 2	-	-	C	-	C	A	-	0.01
Haplotype 3	-	-	C	A	C	A	-	0.18
Haplotype 4	A	T	-	A	C	A	-	0.04
Haplotype 5	A	T	C	-	-	-	0.11	0.08
Haplotype 6	A	T	C	-	C	A	-	0.01
Haplotype 7	A	T	C	A	C	A	-	**0.54**

***** The dashed line (-) in this table indicates that the desired nucleotide is similar to haplotype 1 nucleotide.

**Table 4 animals-12-01423-t004:** Fat metabolism related genes located within the candidate regions in *O. aries* and their orthologous area in *B. taurus*.

Species	Chromosome	RefSeq Number	Gene Name	Gene Symbol	Function
*Ovis aries*	X	NM_001037811	hydroxysteroid (17-beta) dehydrogenase 10	*HSD17B10*	lipid metabolic process

	X	NM_000044	androgen receptor	*Ar*	lipid binding
	X	NM_173963	synaptophysin	*Syp*	lipid binding (Cholestrol binding)
*Bos taurus*	5	NM_002715	protein phosphatase 2 (formerly 2A), catalytic subunit, alpha isoform	*PPP2CA*	cellular lipid metabolic process lipid metabolic process memberan lipid metabolic process sphingolipid metabolic process
	X	NM_001034500	emopamil binding protein (sterol isomerase)	*EBP*	lipid metabolic process

**Table 5 animals-12-01423-t005:** Developmental process or gene expression related genes, located within the candidate regions in *O. aries*.

Chr.	Gene Symbol	Gene Name	Functions *
5	*TCF7*	Transcription factor 7 (T-cell specific, HMG-box)	regulation of gene expression
7	*PTGDR*	Prostaglandin D2 receptor (DP)	developmental process
	*NID2*	Nidogen 2 (osteonidogen)	cellular macromolecule metabolic process
X	*AR*	Androgen receptor	gland development, organ development, system development, anatomical structure development, regulation of gene expression
	*FOXP3*	Forkhead box P3	organ development, system development, anatomical structure development, regulation of developmental process, regulation of gene expression
	*FGD1*	FYVE, RhoGEF and PH domain containing 1	organ development, system development, anatomical structure development, regulation of developmental process
	*BMP15*	Bone morphogenetic proteins 15	organ development, system development, anatomical structure development, regulation of developmental process
	*HSD17B10*	hydroxysteroid (17-beta) dehydrogenase 10	organ development, system development, anatomical structure development
	*TIMP1*	TIMP metallopeptidase inhibitor 1	organ development, system development, anatomical structure development
	*PFKFB1*	Hydroxysteroid (17-beta) dehydrogenase 10	organ development, system development, anatomical structure development
	*SLC35A2*	Solute Carrier Family 35 Member A2	organ development, system development, anatomical structure development
	*ALAS2*	Aminolevulinate, delta-, synthase 2	organ development, system development, anatomical structure development
	*HEPH*	Hephaestin	organ development, system development, anatomical structure development
	*PCSK1N*	Proproteinconvertasesubtilisin/kexin type 1 inhibitor	organ development, system development, anatomical structure development
	*SHROOM4*	Shroom family member 4	organ development, system development, anatomical structure development
	*CACNA1F*	Calcium channel, voltage-dependent, L type, alpha 1F subunit	system development, anatomical structure development
	*TFE3*	Transcription factor binding to IGHM enhancer 3	regulation of developmental process, regulation of gene expression
	*ELK1*	ELK1, member of ETS oncogene family	regulation of gene expression
	*KDM5C*	Lysine (K)-specific demethylase 5C	regulation of gene expression
	*ZNF41, 81*	Zinc finger protein 41, 81	regulation of gene expression

***** gland development: The process whose specific outcome is the progression of a gland over time, from its formation to the mature structure. A gland is an organ specialized for secretion. organ development: Development of a tissue or tissues that work together to perform a specific function or functions. Organs are commonly observed as visibly distinct structures, but may also exist as loosely associated clusters of cells that work together to perform a specific function or functions. system development: The process whose specific outcome is the progression of an organismal system over time, from its formation to the mature structure. A system is a regularly interacting or interdependent group of organs or tissues that work together to carry out a given biological process. anatomical structure development: The biological process whose specific outcome is the progression of an anatomical structure from an initial condition to its mature state. An anatomical structure is any biological entity that occupies space and is distinguished from its surroundings. developmental process: A biological process whose specific outcome is the progression of an integrated living unit: an anatomical structure (which may be a subcellular structure, cell, tissue, or organ), or organism over time from an initial condition to a later condition. regulation of developmental process: Any process that modulates the frequency, rate, or extent of development, the biological process whose specific outcome is the progression of a multicellular organism over time from an initial condition to a later condition. regulation of gene expression: Any process that modulates the frequency, rate, or extent of gene expression. This includes the production of an RNA transcript as well as any processing to produce a mature RNA product or an mRNA and the translation of that mRNA into protein.

## Data Availability

All data generated or analyzed during this study are included in this manuscript and its Appendix A. In addition, more detail data are available from the corresponding author on reasonable request.

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
