# Peer review of "Hitchhiking Mapping of Candidate Regions Associated with Fat Deposition in Iranian Thin and Fat Tail Sheep Breeds Suggests New Insights into Molecular Aspects of Fat Tail Selection"

_animals, 2022, doi:10.3390/ani12111423_

Round 1
Reviewer 1 Report
This a well written article, very interesting and within the scope of the Journal.
Author Response
Author's Notes
We would like to thank you for giving us the opportunity to revise and improve our manuscript. We have provided a systemic response to each question/comment and alterations on a point by point basis as appended below. The changes within the main text had been shown with red color. Hopefully the changes have addressed the concerns.
================== Reviewer 1
Comments and Suggestions for Authors:
This is a well written article, very interesting and within the scope of the Journal.
- Answer: The authors deeply appreciate the Reviewer for the time spending for studying our manuscript and for the positive feedback.
Reviewer 2 Report
Line 17: “to suggests” change for infinitive
Line 18: rephrase “study was novel in the sense that most of the exploratory genome scan studies”, since there were such former studies
Line 21: it needs explanation because there are three sheep groups: fat-tailed, fat-rumped and without intense fat store at this body part
Line 67-68-69: revise this sentence “However, a standard QTL mapping approach requires the segregating populations to be established for the association between the phenotype and genotype can be established”
Line 75-76-77: revise this sentence because it is questionable from professional point of view: “This approach is especially suitable for the traits like fat tail where the phenotypic records in almost all countries grazing these breeds are sparse and their measurement is difficult.” How was this trait selected in the millennia before molecular genetics?
Line 118-119: The authors specify exactly which anatomical body part of the animal the "fat deposition" in the study refers to! “In this study, three regions recognized as candidate regions associated with fat deposition were chosen for further analysis.”
Sub-chapter “2.1. Animal sampling and genotyping”: The authors give a clear picture of their own database!
Line 135: What is the significance of the degree of relatedness when matching phenotype and genotype?
Line 174: maybe you can choose another way to refer a literature
Line 198-201: this paragraph is fuzzy: How do you calculate effective population size if you consider males to be hemizygous?
Figure 2: correct the legend
Take care of genes PTGDR and HSD17B10, maybe they needs corrections (here and it is worth to do generally). The gene HSD15B is presented in Abstract and Discussion only. If it is really important, mention it in the Results.
The shortcoming of the manuscript is that the authors do not emphasize the candidate genes with functions based on own investigation, only regions and sites, and works of others.
The gene for fat-tail characteristics has not been identified by the authors. It is not clear whether the increase in homozygosity of the studied regions is due to selection for the fatty tail or selection for another traits. Authors have to perform a more convincing wording.
The term “haplotype block” is appearing in the Discussion only.(?). A massive re-edition is obligatory.
The term “ancestral allele” requires more description to know the deviation/sameness of investigated breeds.
Author Response
Author's Notes
We would like to thank you for giving us the opportunity to revise and improve our manuscript. We have provided a systemic response to each question/comment and alterations on a point by point basis as appended below. The changes within the main text had been shown with red color. Hopefully the changes have addressed the concerns.
=================== Reviewer 2
Comments and Suggestions for Authors:
Comment 1: Line 17: “to suggests” change for infinitive.
Response 1: The text was altered as the referee mentioned.
Comment 2: Line 18: rephrase “study was novel in the sense that most of the exploratory genome scan studies”, since there were such former studies.
Response 2: Thank you for the comment. The text was altered to the referee suggested rephrasing.
Comment 3: Line 21: it needs explanation because there are three sheep groups: fat-tailed, fat-rumped and without intense fat store at this body part.
Response 3: The text was revised to address the referee's concern, as all Iranian sheep genetic resources can be divided into only fat tail and thin tailed sheep breeds.
Comment 4: Line 67-68-69: revise this sentence “However, a standard QTL mapping approach requires the segregating populations to be established for the association between the phenotype and genotype can be established”.
Response 4: The text revised as the referee suggested to " QTL mapping basically relies on detecting correlations between genetic markers and phenotypic traits in a segregating population"
Comment 5: Line 75-76-77: revise this sentence because it is questionable from professional point of view: “This approach is especially suitable for the traits like fat tail where the phenotypic records in almost all countries grazing these breeds are sparse and their measurement is difficult.” How was this trait selected in the millennia before molecular genetics?
Response 5: The text was revised as below, to address the referee's concern.
" This approach is especially suitable for the traits like sheep tail pattern where all sheep breeds can be easily classified in different classes (e.g. thin and fat tailed sheep groups) and then, offer ideal materials for comparative analysis of their genetic basis."
Comment 6: Line 118-119: The authors specify exactly which anatomical body part of the animal the "fat deposition" in the study refers to! “In this study, three regions recognized as candidate regions associated with fat deposition were chosen for further analysis.”
Response 6: The text was altered as the referee suggested.
Comment 7: Sub-chapter “2.1. Animal sampling and genotyping”: The authors give a clear picture of their own database!
Response 7: The authors thank the Reviewer again for the valuable comments that considerably improved our manuscript.
Comment 8: Line 135: What is the significance of the degree of relatedness when matching phenotype and genotype?
Response 8: Thank you for the comments. Actually, since we have previously assumed that this study may also identify some other regions that may explain differences between fat and thin tailed sheep breeds for production, reproduction, and survivability (in addition to fat deposition, that aimed in the current study), therefore, we have used different scenario to direct us to the signatures that are more likely associated with fat deposition. For instance, two factors were considered for selecting the samples: selection of unrelated animals and sampling those that spanned the diversity of the breed. Therefore, the degree of relatedness has been used in this step to cover as much diversity of animals as possible in the collected samples.
Comment 9: Line 174: maybe you can choose another way to refer a literature.
Response 9: Thank you for the comment, the referring to the literature was altered.
Comment 10: Line 198-201: this paragraph is fuzzy: How do you calculate effective population size if you consider males to be hemizygous?
Response 10: Actually we did not calculate effective population size in the current study, but it is mentioned that the analysis for chromosome X were performed by using only females (In addition to the analysis with all animals, which was accompanied by similar results). This was performed because some published studied suggested this, mostly due to different effective population sizes of males and females for sexual and autosomal chromosomes (Sayres, 2008).
- Sayres, M.A.W. Genetic diversity on the sex chromosomes. Genome biology and evolution 2018, 10, 1064
Comment 11: Figure 2: correct the legend.
Response 11: The legend was altered as the referee suggested.
Comment 12: Take care of genes PTGDR and HSD17B10, maybe they needs corrections (here and it is worth to do generally). The gene HSD15B is presented in Abstract and Discussion only. If it is really important, mention it in the Results.
Response 12: The authors thank the referee for the valuable comment, Regarding PTGDR and HSD17B10 genes: The PTGDR gene encodes a member of protein that is reported to be a receptor for prostaglandin D2, which is a lipid mediator of autocrine and paracrine functions (García-Solaesa et al., 2014) and HSD17B10 gene provides instructions for making a protein called HSD10. This protein is located within mitochondria, where it has several different functions specially in the pathway of fatty acid beta-oxidation, which is part of lipid metabolism (Oerum et al., 2017). Both genes have also a variety of roles in developmental process of organs/glands (Dennis et al., 2003). In total, their multiple effect on lipid metabolisms and organ/gland development have been considered to suggest them as candidate genes for fat deposition in sheep.
The gene of HSD17B10 has been meant in these sections, therefore the text for Abstract and Discussion was corrected to address the referee's point.
- García-Solaesa, V.; Sanz-Lozano, C.; Padrón-Morales, J.; Hernández-Hernández, L.; García-Sánchez, A.; Rivera-Reigada, M.L.; Dávila-González, I.; Lorente-Toledano, F.; Isidoro-García, M. The prostaglandin D2 receptor (PTGDR) gene in asthma and allergic diseases. Allergologia et immunopathologia 2014, 42, 64–68.
- Oerum, S.; Roovers, M.; Leichsenring, M.; Acquaviva-Bourdain, C.; Beermann, F.; Gemperle-Britschgi, C.; Fouilhoux, A.; Korwitz-Reichelt, A.; Bailey, H.J.; Droogmans, L.; et al. Novel patient missense mutations in the HSD17B10 gene affect dehydrogenase and mitochondrial tRNA modification functions of the encoded protein. Biochimica et Biophysica Acta (BBA) - Molecular Basis of Disease 2017, 1863, 3294–3302.
- Dennis, G.; Sherman, B.T.; Hosack, D.A.; Yang, J.; Gao, W.; Lane, H.C.; Lempicki, R.A. DAVID: database for annotation, visualization, and integrated discovery. Genome biology 2003, 4, 1–11.
Comment 13: The shortcoming of the manuscript is that the authors do not emphasize the candidate genes with functions based on own investigation, only regions and sites, and works of others.
Response 13: While the authors thank the Reviwer's comment, actually the regions of interest and one of the genes identified in our study confirmed by our investigations: Firstly: the regions have been selected for fine mapping in the current study that previously detected in our own whole genome analysis using two independent experiments, including Iranian, and ovine HapMap genotyping data contrasting thin and fat tailed breeds (distributed all around the world). These regions have been also confirmed in two ways in independent studies:
1- Same breeds, different animals (Zare et al., 2016)
2- Different breeds, different animals (Kang et al., 2017; Ma et al., 2018; Mastrangelo et al., 2018; Ahbara et al., 2019).
And secondly: One of the main genes that have been suggested in our study (PPP2CA gene), has been also investigated by our team, in an independent dataset collected from 140 Zel sheep, and 165 Lori-Bakhtiari sheep breeds for confirming its association with fat deposition (Zare et al., 2016, Supervised by Dr. Mohammad Moradi-Shahrebabak). Two patterns were identified in this study and the results of sequencing showed that in Lori-Bakhtiari, Del/Del genotype resulted in heavier fat tail than T/T genotype (5.20±0.21kg vs 3.28±0.12kg) (P<0.05) while, in Zel, the effect of genotypes on carcass fat percentage and triglyceride was significant, so that the T/T genotype had more carcass fat percentage comparing to Del/Del genotypes (P<0.05). This study has been referred in the Discssion (Line 567-576). However, the authors of this article agree that more genes require to be confirm in independent studies.
Comment 14: The gene for fat-tail characteristics has not been identified by the authors. It is not clear whether the increase in homozygosity of the studied regions is due to selection for the fatty tail or selection for another traits. Authors have to perform a more convincing wording.
Response 14: Referee 2 stated that the observed selective sweeps could be related to other traits or may be spurious, while we agree to this point but as described earlier, we have addressed this in the following ways and discussed by details in our previous article in BMC Genetics journal (Moradi et al., 2012):
- Two Iranian breeds were chosen for this study, it was felt this geographic proximity would reduce extraneous differences between the breeds due to factors such as climate, disease and pasture types. And on the other hand, we tried to cover as much diversity of animals as possible in the collected samples, so that the main difference between samples would be due to fat deposition in the tail.
- However, we expected that some signals would still be related to other traits such as production, reproduction and survivability as these traits are somewhat different in these Iranian breeds, so we then independently validated the results using a different set of fat and thin tail breeds, distributed all around the world, obtained from the Ovine HapMap project.
- A variety of alternative comparisons, with the various thin and fat tail breeds using the combined HapMap and Zel-Lori Bakhtiari data set were also performed, in addition to the comparisons presented in Moradi et al. (2012). Comparison of different breeds with different characteristics revealed that the three regions used in the current study, would still be candidate regions for fat tail deposition trait.
- We also believe that more detailed and larger scale experiments are needed to independently confirm the results reported in this study.
Comment 15: The term “haplotype block” is appearing in the Discussion only.(?). A massive re-edition is obligatory.
Response 15: While the authors thank again the referee for the valuable comments, actually this analysis has been performed to discuss the results of |iHS| statistics, that is why we reported the results of "haplotype block" only in the Discussion. However, the method of "haplotype block" analysis was added to "Materials and Methods" section (Line 283) to address the referee's comment.
Comment 16: The term “ancestral allele” requires more description to know the deviation/sameness of investigated breeds.
Response 16: Thank you for the comment. The derived allele is the one which appears to have arisen within sheep populations, while the ancestral allele is the one present during speciation of sheep. Alleles having diverged through mutation are called derived alleles (DA), while alleles that persist in their initial state are termed ancestral alleles (AA) (Fay et al, 2000). A reasonable method to assess AA is by comparing shared polymorphic sites of closely related species. Alleles that are still intact and shared by all the related species are most likely the ancestral allele (Rocha et al., 2014). These definitions can be found in the literature we have referred to (eg. Voight et al., 2006; McRae 2012).
- Fay, J.C.; Wu, C.-I. Hitchhiking under positive Darwinian selection. Genetics 2000, 155, 1405–1413.
- Rocha, D.; Billerey, C.; Samson, F.; Boichard, D.; Boussaha, M. Identification of the putative ancestral allele of bovine single‐nucleotide polymorphisms. Journal of Animal Breeding and Genetics 2014, 131, 483–486.
- Voight, B.F.; Kudaravalli, S.; Wen, X.; Pritchard, J.K. A map of recent positive selection in the human genome. PLoS biology 2006, 4, e72.
- McRae, K.M. Signatures of selective sweeps in parasite selection flocks 2012. University of Otago, New Zealand
Reviewer 3 Report
The current work aimed at the studying the patterns of variation in previously identified candidate regions associated with fat deposition in thin tail (Zel) and fat tail (Lori-Bakhtiari) sheep breeds. Authors performed the genotyping of SNPs in the three candidate regions at autosomes 5, 7 and X using a Sequenom Mass ARRAY platform. Based on research results the signature of selection in studied regions were confirmed.
I found the study very interesting. The manuscript is well structured, easy to read, the results are well described and discussed.
I believe, that the manuscript can be accepted for publication.
Author Response
Author's Notes
We would like to thank you for giving us the opportunity to revise and improve our manuscript. We have provided a systemic response to each question/comment and alterations on a point by point basis as appended below. The changes within the main text had been shown with red color. Hopefully the changes have addressed the concerns.
=================== Reviewer 3
Comments and Suggestions for Authors:
- The current work aimed at the studying the patterns of variation in previously identified candidate regions associated with fat deposition in thin tail (Zel) and fat tail (Lori-Bakhtiari) sheep breeds. Authors performed the genotyping of SNPs in the three candidate regions at autosomes 5, 7 and X using a Sequenom Mass ARRAY platform. Based on research results the signature of selection in studied regions were confirmed. I found the study very interesting. The manuscript is well structured, easy to read, the results are well described and discussed.
- I believe, that the manuscript can be accepted for publication.
- The authors deeply appreciate the Reviewer for the positive feedback and for giving us the opportunity to revise and improve our manuscript. Thank you very much for your kind consideration.
Reviewer 4 Report
The manuscript is generally properly written and contains some novel information. Based on the previous results, the author carried out in-depth analysis and obtained meaningful results.The introduction is enough described to understand the subject of study. Material and methods are properly described and consist of details, which are necessary it this type of article.Result and discussion are properly and clearly described. However, it requires a few improvements.
1.It is worth adding research progress of fat and thin tail sheep in Introduction. It will be easier for the reader to understand the differences.
2.The number of sheep breeds is inconsistent. For example, line 134, 45 samples, 37 females. However, line 201, 36 female. And, there were 47 samplesin the author's BMC genetics article in 2012.
3.Line 277, the reference genome is not up-to-date.
4.Lines 331-333, Are the values of autosomal and X chromosome calculated from Zel-Lori Bakhtiari date in this manuscript? If not, please provide the source of the reference.
5.Line 535: State error, ARS-UI_Ramb_v2.0 is the latest sheep reference genome.
6.Are the Zel-Lori Bakhtiari data published in the public database?
Author Response
We would like to thank you for giving us the opportunity to revise and improve our manuscript. We have provided a systemic response to each question/comment and alterations on a point by point basis as appended below. The changes within the main text had been shown with red color. Hopefully the changes have addressed the concerns.
=================== Reviewer 4
Comments and Suggestions for Authors:
The manuscript is generally properly written and contains some novel information. Based on the previous results, the author carried out in-depth analysis and obtained meaningful results. The introduction is enough described to understand the subject of study. Material and methods are properly described and consist of details, which are necessary it this type of article. Result and discussion are properly and clearly described. However, it requires a few improvements.
Comment 1: It is worth adding research progress of fat and thin tail sheep in Introduction. It will be easier for the reader to understand the differences.
Response 1: Thank you for the comment. Actually, a brief overview on the genomic outcomes, including proposed potential genes generated from investigations on the sheep tail phenotype has been reported by Kalds et al. (2021) in Animal Genetics journal (from the first study conducted to identify genomic regions associated with fat deposition by our team (Moradi et al., 2012), up to now), however, the following brief description on the research progress in this area, has been added in the Introduction, as the referee suggested.
- Various genomic-based studies have been conducted to understand the genetic basis and genomic architecture of sheep tails and to find the specific causal genomic variant(s) contributing to sheep tail pattern making (Moradi et al. 2012; Moioli et al. 2015; Wei et al. 2015; Pan et al. 2019; Dong et al. 2020; Li et al. 2020; Shao et al. 2020; Luo et al. 2021). A brief overview on genomic outcomes, including proposed potential genes generated from investigations on the sheep tail phenotype has been reported by Kalds et al. (2021). Additionally, transcriptomic analyses were performed, providing sets of potential genes that contribute to the formation and the biological emergence of sheep tails (Wang et al. 2014; Ma et al. 2018; Bakhtiarizadeh & Alamouti, 2020; Zhang et al. 2021). Taken together, these research efforts have revealed several high-ranking candidate genes with no current consistency or solid opinion about their variant causalities and expression nature.
- Moradi, M.H.; Nejati-Javaremi, A.; Moradi-Shahrbabak, M.; Dodds, K.G.; McEwan, J.C. Genomic scan of selective sweeps in thin and fat tail sheep breeds for identifying of candidate regions associated with fat deposition. BMC Genetics 2012, 10, 13.
- Dimauro, C.; Nicoloso, L.; Cellesi, M.; Macciotta, N.P.P.; Ciani, E.; Moioli, B.; Pilla, F.; Crepaldi, P. Selection of discriminant SNP markers for breed and geographic assignment of Italian sheep. Small Ruminant Research 2015, 128, 27-33.
- Wei, C.; Wang, H.; Liu, G.; Wu, M.; Cao, J.; Liu, Z.; Liu, R.; Zhao, F.; Zhang, L.; Lu, J. Genome-wide analysis reveals population structure and selection in Chinese indigenous sheep breeds. BMC genomics 2015, 16, 1–12.
- Pan, Z.; Li, S.; Liu, Q.; Wang, Z.; Zhou, Z.; Di, R.; An, X.; Miao, B.; Wang, X.; Hu, W. Rapid evolution of a retro-transposable hotspot of ovine genome underlies the alteration of BMP2 expression and development of fat tails. BMC genomics 2019, 20, 1–15.
- Dong, K.; Yang, M.; Han, J.; Ma, Q.; Han, J.; Song, Z.; Luosang, C.; Gorkhali, N.A.; Yang, B.; He, X.; et al. Genomic analysis of worldwide sheep breeds reveals PDGFD as a major target of fat-tail selection in sheep. BMC Genomics 2020, 21, 800.
- Li, X.; Yang, J.; Shen, M.; Xie, X.-L.; Liu, G.-J.; Xu, Y.-X.; Lv, F.-H.; Yang, H.; Yang, Y.-L.; Liu, C.-B.; et al. Whole-genome resequencing of wild and domestic sheep identifies genes associated with morphological and agronomic traits. Nature Communications 2020, 11, 2815.
- Shao, J.; He, S.; Pan, X.; Yang, Z.; Nanaei, H.A.; Chen, L.; Li, R.; Wang, Y.; Gao, S.; Xu, H. Allele-specific expression reveals the phenotypic differences between thin-and fat-tailed sheep. 2020.
- Luo, R.; Zhang, X.; Wang, L.; Zhang, L.; Li, G.; Zheng, Z. GLIS1, a potential candidate gene affect fat deposition in sheep tail. Molecular biology reports 2021, 48, 4925–4931.
- Kalds, P.; Luo, Q.; Sun, K.; Zhou, S.; Chen, Y.; Wang, X. Trends towards revealing the genetic architecture of sheep tail patterning: Promising genes and investigatory pathways. Animal Genetics 2021, 52, 799–812.
- Wang, X.; Zhou, G.; Xu, X.; Geng, R.; Zhou, J.; Yang, Y.; Yang, Z.; Chen, Y. Transcriptome profile analysis of adipose tissues from fat and short-tailed sheep. Gene 2014, 549, 252–257.
- Ma, L.; Li, Z.; Cai, Y.; Xu, H.; Yang, R.; Lan, X. Genetic variants in fat‐and short‐tailed sheep from high‐throughput RNA‐sequencing data. Animal genetics 2018, 49, 483–487.
- Bakhtiarizadeh, M.R.; Alamouti, A.A. RNA-Seq based genetic variant discovery provides new insights into controlling fat deposition in the tail of sheep. Scientific Reports 2020, 10, 1–13.
Comment 2: The number of sheep breeds is inconsistent. For example, line 134, 45 samples, 37 females. However, line 201, 36 females. And, there were 47 samples in the author's BMC genetics article in 2012.
Response 2: Thank you very much for the comment. Actually 47 animals per each breed has been initially genotyped and as described in details in our previous article in BMC Genetics, two animals per each breed were excluded due to quality control steps (Animal call rate and PCA plot), that is why we have used 45 samples (including 37 females per each breed) in the current study. The inconsistency in 134 and 201 lines was a mistake and has been corrected.
Comment 3: Line 277, the reference genome is not up-to-date.
Response 3: Thank you very much for the comment. The text was altered to address the referee's point. Actually OAR v3.1 reference genome has been widely used in almost all of association studies performed for fat deposition in sheep (e.g. Moioli et al. 2015; Wei et al. 2015; Yuan et al. 2016; Dong et al. 2020; Zhao et al., 2020; Zhu et al. 2021), therefore, this version has been also reported in the current study due to easier comparison of our obtained results to previously published studies (although the latest version of OAR v.4 has been also considered and no special differences was observed).
- Moioli, B.; Pilla, F.; Ciani, E. Signatures of selection identify loci associated with fat tail in sheep. Journal of animal science 2015, 93, 4660–4669.
- Wei, C.; Wang, H.; Liu, G.; Wu, M.; Cao, J.; Liu, Z.; Liu, R.; Zhao, F.; Zhang, L.; Lu, J. Genome-wide analysis reveals population structure and selection in Chinese indigenous sheep breeds. BMC genomics 2015, 16, 1–12.
- Yuan, Z.; Liu, E.; Liu, Z.; Kijas, J.W.; Zhu, C.; Hu, S.; Ma, X.; Zhang, L.; Du, L.; Wang, H. Selection signature analysis reveals genes associated with tail type in Chinese indigenous sheep. Animal genetics 2017, 48, 55–66.
- Dong, K.; Yang, M.; Han, J.; Ma, Q.; Han, J.; Song, Z.; Luosang, C.; Gorkhali, N.A.; Yang, B.; He, X.; et al. Genomic analysis of worldwide sheep breeds reveals PDGFD as a major target of fat-tail selection in sheep. BMC Genomics 2020, 21, 800.
- Zhao, F.; Deng, T.; Shi, L.; Wang, W.; Zhang, Q.; Du, L.; Wang, L. Genomic scan for selection signature reveals fat deposition in Chinese indigenous sheep with extreme tail types. Animals 2020, 10, 773.
- Zhu, C.; Li, N.; Cheng, H.; Ma, Y. Genome wide association study for the identification of genes associated with tail fat deposition in Chinese sheep breeds. Biology open 2021, 10, bio054932.
Comment 4: Lines 331-333, Are the values of autosomal and X chromosome calculated from Zel-Lori Bakhtiari date in this manuscript? If not, please provide the source of the reference.
Response 4: Yes, the values of autosomal and X chromosome calculated in the current manuscript.
Comment 5: Line 535: State error, ARS-UI_Ramb_v2.0 is the latest sheep reference genome.
Response 5: Thank you for the comment. Actually, since we aimed to determine the biological functions of each gene reported in our region of interest using functional annotation bioinformatics tools of DAVID and this tool is not working with Ramb_v2 reference genome yet, Texel_OAR_v3.1 reference genome has been used in the current study, as it has been widely used in different studies for functional analysis in sheep (Ahbara et al., 2019; Alvarez et al., 2020; Rekik et al., 2022).
- Ahbara, A.; Bahbahani, H.; Almathen, F.; Al Abri, M.; Agoub, M.O.; Abeba, A.; Kebede, A.; Musa, H.H.; Mastrangelo, S.; Pilla, F. Genome-wide variation, candidate regions and genes associated with fat deposition and tail morphology in Ethiopian indigenous sheep. Frontiers in genetics 2019, 9, 699.
- Álvarez, I.; Fernández, I.; Traoré, A.; Pérez-Pardal, L.; Menéndez-Arias, N.A.; Goyache, F. Ancient Homozygosity Segments in West African Djallonké Sheep Inform on the Genomic Impact of Livestock Adaptation to the Environment. Animals 2020, 10.
- Rekik, E.; Ahbara, A.M.; Abate, Z.; Goshme, S.; Getachew, T.; Haile, A.; Rischkowsky, B.; Mwacharo, J.M. Genomic analysis of 10 years of artificial selection in community‐based breeding programs in two Ethiopian indigenous sheep breeds. Animal Genetics 2022, 00, 1-5.
Comment 6: Are the Zel-Lori Bakhtiari data published in the public database?
Response 6: All data generated or analyzed during this study are included in this manuscript and its supplementary information files. In addition, all genotyping data of Zel-Lori Bakhtiari would be available on the public databases suggest by Animals journal.
Round 2
Reviewer 2 Report
The manuscript is accepted in present form.